# Diagnosis of *Pneumocystis jirovecii* Pneumonia in Pediatric Patients in Serbia, Greece, and Romania. Current Status and Challenges for Collaboration

**DOI:** 10.3390/jof6020049

**Published:** 2020-04-17

**Authors:** Valentina Arsić Arsenijevic, Timoleon-Achilleas Vyzantiadis, Mihai Mares, Suzana Otasevic, Athanasios Tragiannidis, Dragana Janic

**Affiliations:** 1National Reference Laboratory for Medical Mycology, Institute of Microbiology and Immunology, Faculty of Medicine, University of Belgrade, 11000 Belgrade, Serbia; 2First Department of Microbiology, School of Medicine, Aristotle University of Thessaloniki, 541 24 Thessaloniki, Greece; tavyz@auth.gr; 3Laboratory of Antimicrobial Chemotherapy, Ion Ionescu de la Brad University, 700490 Iasi, Romania; mycomedica@gmail.com; 4Department of Microbiology & Public Health Institute Clinical Center of Nis, Faculty of Medicine, University of Nis, 18000 Nis, Serbia; otasevicsuzana@gmail.com; 5Haematology Oncology Unit, Second Department of Pediatrics, School of Medicine, Aristotle University of Thessaloniki, 541 24 Thessaloniki, Greece; atragian@auth.gr; 6Institute for Oncology and Radiology of Serbia, Faculty of Medicine, University of Belgrade, 11000 Belgrade, Serbia; dragana.janic@udk.bg.ac.rs

**Keywords:** *Pneumocystis jirovecii* pneumonia, pediatric patients, molecular diagnosis, Serbia, Greece, Romania

## Abstract

*Pneumocystis jirovecii* can cause fatal *Pneumocystis* pneumonia (PcP). Many children have been exposed to the fungus and are colonized in early age, while some individuals at high risk for fungal infections may develop PcP, a disease that is difficult to diagnose. Insufficient laboratory availability, lack of knowledge, and local epidemiology gaps make the problem more serious. Traditionally, the diagnosis is based on microscopic visualization of *Pneumocystis* in respiratory specimens. The molecular diagnosis is important but not widely used. The aim of this study was to collect initial indicative data from Serbia, Greece, and Romania concerning pediatric patients with suspected PcP in order to: find the key underlying diseases, determine current clinical and laboratory practices, and try to propose an integrative future molecular perspective based on regional collaboration. Data were collected by the search of literature and the use of an online questionnaire, filled by relevant scientists specialized in the field. All three countries presented similar clinical practices in terms of PcP prophylaxis and clinical suspicion. In Serbia and Greece the hematology/oncology diseases are the main risks, while in Romania HIV infection is an additional risk. Molecular diagnosis is available only in Greece. PcP seems to be under-diagnosed and regional collaboration in the field of laboratory diagnosis with an emphasis on molecular approaches may help to cover the gaps and improve the practices.

## 1. Introduction

*Pneumocystis jirovecii* (formerly known as *Pneumocystis carinii*) is a fungal pathogen that can cause fatal *Pneumocystis* pneumonia (PcP) [1]. *Pneumocystis* was first described in 1909. It was initially identified as protozoa, but the analysis of the nucleic acid composition and mitochondrial DNA identified the organism as a unicellular fungus [2,3]. *P. jirovecii* was described as a cause of interstitial pneumonia in severely malnourished and premature infants during World War II in Central and Eastern Europe [4]. Before the HIV pandemia, infection with *P. jirovecii* was sporadic. Since the 1980s it has become the most common life-threatening opportunistic infection in persons with HIV, with over 100,000 PcP cases reported in the first decade of the HIV epidemic in the United States [5]. Other immunocompromised patients are at increased risk for PcP, such as transplant recipients, patients with hematologic and solid malignancies, and patients receiving immuno-modulatory therapies or with pre-existing chronic lung conditions.

Once inhaled, the trophic form of *P. jirovecii* attaches to the alveoli, colonizing lungs. Most children exposed to *P. jirovecii* had been colonized in early age, usually without symptoms [6]. PcP develops due to uncontrolled replication of *P. jirovecii* [7] when cellular and humoral immunity fails to control its replication. The impaired host immunity allows *P. jirovecii* replication and development of PcP, a usually-serious pneumonia [8] with non-specific symptoms [9]. As *P. jirovecii* is found in the lungs of healthy individuals, it can be involved in hospital outbreaks too [10].

Due to non-specific pulmonary symptoms and signs, PcP is difficult to diagnose and appears to be easily under-diagnosed and under-estimated. In addition, use of prophylaxis, insufficient laboratory availability, and local epidemiology gaps make the problem more difficult, especially in the pediatric hematology-oncology (PHO) population [10], including patients after hematopoietic stem cell transplantation (HSCT) [11]. The rising number of immunocompromised children indicates the need for surveillance and better opportunities for diagnosing PcP which may have a more sub-acute course in pediatrics. Pulmonary symptoms are non-specific, and other findings may include tachypnea, fever, or tachycardia, while the extrapulmonary manifestations are rare. Additional findings in children with severe disease include cyanosis, nasal flaring, and intercostal retractions. Pulmonary examination may reveal mild crackles in bronchi or normal findings in even half of the patients [12]. The chest radiographic findings are important, but may be normal in patients with mild disease, so the normal chest radiography findings alone do not rule out PcP. In most patients with PcP, diffuse bilateral infiltrates, extending from the perihilar region, may be visible. High-resolution computed tomography scanning of chest is useful and the typical appearance shows patchy areas of ground-glass attenuation with a background of interlobular septal thickening [13].

Traditionally, the confirmation of PcP diagnosis is based on microscopic visualization of *P. jirovecii* in respiratory specimens, but due to low sensitivity of the microscopy/histology approach, molecular diagnostics have been developed with growing importance. The aim of the study was to collect initial indicative data from Serbia, Greece, and Romania concerning pediatric patients with suspected PcP in order to: (i) find the key underlying diseases and risks, (ii) determine current clinical and laboratory practices, and iii) propose an integrative development of molecular diagnosis in the basis of increasing regional-scale collaboration.

## 2. Methods

This work was designed as a pilot retrospective analysis that aimed to obtain initial indicative data about clinical and laboratory scenarios when PcP is suspected in hospitalized pediatric immunocompromised patients [namely hematology-oncology, hematopoietic stem cell transplantation, HIV+, cystic fibrosis (CF), and organ transplant (OT) recipients] in Serbia, Greece, and Romania. The data were mostly collected by the use of personal communication and an online questionnaire, circulated between several laboratories and clinical departments, which contained questions about number of patients and types of underlying diseases, co-morbidity, prophylaxis/treatment applied, laboratory assays used for PcP diagnosis, and outcome, if this information existed in their archives. A PubMed and Scopus literature search containing terms such as “*Pneumocystis* pneumonia“, “children“, “pediatric patients“, “Serbia“, “Romania“ or “Greece“ for the last twenty years was also carried out.

## 3. Results

### 3.1. Data from Serbia

PcP was included in the assessment of serious fungal infections in Serbia and was estimated in 60 new cases per year in the whole population [14]. However, published pediatric PcP cases are absent, and the survey on suspected pediatric PcP occurrence was done primarily in settings of PcP prophylaxis. In Serbia, PHO patients are treated in five centers; one center for brain, bone and retinoblastoma cases, and four centers for leukemias, including acute lymphoblastic leukemia (ALL), lymphomas, and solid tumors. Additionally, there is one bone marrow transplant unit (BMT) performing pediatric HSCT. In the country there is only one kidney transplant unit for the pediatric population, while other organ transplantations are mostly done abroad. Patients with CF are almost exclusively treated at the pulmonary ward of one tertiary care pediatric hospital, and an adult infectious diseases department has a ward treating all HIV-infected children (Table 1). Due to the lack of either the 1, 3-β-D-glucan test (BDG) or *Pneumocystis* PCR for diagnosis of PcP, the PcP diagnosis is made exclusively on clinical features. Moreover, identifying *P. jirovecii* in respiratory specimens is not a common practice, despite the fact that several laboratories can perform microscopy or histology.

Two PHO centers, treating leukemias, lymphomas, and solid tumors, reported 20 patients in the last ten years (all of whom suffered from ALL) and the BMT unit reported 28 patients in the same period (Table 1). All patients treated with therapeutic dose of trimethoprim/sulfamethoxazole (TMP-SMX) (20 mg/kg/day) were concomitantly treated with at least one broad-spectrum antibiotic and some of them also with antifungals. However, we were unable to obtain accurate data on mortality of reported patients related to the infectious episode with suspected PcP. There is still no PHO registry in Serbia, but all children with ALL are included in international clinical trials [15]. According to these records and hospital archives, the number of new PHO malignancies in patients aged from 0 to 18 years is estimated at 240 cases per year (including an average of 50 ALL and 30 HSCT cases per year) (Table 1). In these two settings, five PcP cases were suspected per year. All children with ALL and HSCT were given PcP prophylaxis on three consecutive days each week, as required by the protocol. No suspected cases of PcP in the period of 10 years were reported from the centers treating pediatric patients with retinoblastoma, brain and bone tumors, kidney transplant, CF, or HIV.

### 3.2. Data from Romania

Literature data from a single-center study in Romania showed the presence of PcP in infants under six months of age. PcP represented 1.5/1000 hospitalized children, and the reported risk factors were low birth weight, low weight for age, prolonged hospital stay, and HIV infection [16]. The general population at risk of serious fungal infections, including PcP, was also estimated. [17].

Currently in Romania, twelve PHO units treat patients at high risk for PcP but only few hospital laboratories deal with samples for PcP diagnostics. The disease is suspected, diagnosed, and treated mostly based on clinical and radiological findings. However, diagnostic tests like direct microscopy and direct immunofluorescence (DIF) on induced sputum or bronchoalveolar lavage (BAL) are performed sporadically in specific circumstances, but the exact information was not possible to obtain. The main categories of pediatric patients at risk for PcP in Romania are children suffering from ALL and other malignant tumors, HIV positive late-presenters (vertical transmission), or malnourished individuals. Very few cases of suspected PcP were described in CF patients and in children suffering from various innate immunodeficiencies [18]. There are about 5000 cancer cases in children, with 500 new cases diagnosed every year (Table 1).

According to the literature, among PHO patients, the dominant disease is leukemia (25%–27%), followed by lymphomas (15%–17%) [19]. The incidence of PcP in the absence of prophylaxis can vary from 5%–15% in children with HSCT and 22%–45% in patients with ALL [20]. In Romania, in 2018, the incidence of HIV infection in children was 0.13 cases per 100,000. The materno-fetal transmission, accounting for 4.7% of new cases in 2010 had decreased to 1% in 2018 [21]. Despite the antiretroviral (ARV) prophylaxis in newborns with seropositive mothers (219 under prophylaxis out of 220 newborns in 2018), a problem still exists in the case of late presenters with infected but not diagnosed mothers (seven new cases in 2018). Their newborns are at high risk for PcP if specific prophylaxis does not start at the latest after 6 weeks of life and this is the explanation for the quasi-constant number of PcP cases occurring annually in such populations.

### 3.3. Data from Greece

The general population at risk for serious fungal infections, including PcP, has been estimated in Greece [22], but published PcP cases exist mainly for adults [23]. Currently in Greece, there are seven PHO units, one pediatric BMT unit, two CF units exclusively for pediatric cases, and two pediatric HIV units. Solid organ transplantations are performed in adult departments (Table 1).

Diagnosis of PcP is based in the combination of clinical and laboratory data. Microscopy (mainly DIF) is a basic approach, while molecular diagnosis and the serum BDG test are also available. Almost every child at risk for PcP is already under prophylactic treatment with TMP-SMX. Pediatric cases at risk are mostly PHO and transplanted patients.

Data from a reference laboratory in Northern Greece is herein provided (Table 2) as a typical example of constantly low numbers of suspected cases referred and tested (with all available methods) for PcP, In the same period of time the referred adult cases, at the same laboratory, were more than 120 and at least 30 of these were found to be positive. The main reason for the low demand in pediatric patients at risk is that they are a more homogenous and low-numbered group that are followed up by specific protocols and almost all of them are already under relevant prophylaxis, as mentioned above, while this is not often the case in adult population.

## 4. Discussion and Conclusions

It is well known that pediatric patients at risk for PcP include those with lymphoid malignancies, primarily those with ALL as well as HSCT recipients, those suffering from non-Hodgkin lymphoma, those being treated with corticosteroids, and patients with severe lymphocytopenia [24]. Craiu et al. reported PcP in hospitalized infants under six months of age [16]. Larsen et al, showed that in 422 hospitalized infants with acute respiratory tract infection, *P*. *jirovecii* was detected in 16% cases by using a real-time PCR assay. The lowest percentage of positive PCR assay was in infants younger than 49 days (2%), followed by infants of 113 to 265 (13%), and 50 to 112 days old (48%) [25]. Vargas et al, showed positive *P*. *jirovecii* DNA in 51.7% of infants who died unexpectedly in the community, while only 15% of them had pneumonia. *P*. *jirovecii* infection was also more frequent than viral infection before the age of 6 months, and it showed a consistent peak between 2 and 5 months of age [26]. A high prevalence of *P*. *jirovecii* infection in preterm neonates was detected in Spain, and was associated with high risk of developing neonatal respiratory distress syndrome [27]. 

Based on this first concise survey that tried to provide information concerning the state of PcP infection in the pediatric population in our region, limited data was found for PcP in neonates or infants. In Serbia, Greece, and Romania PcP is suspected mainly in pediatric population with ALL and HCST. Although HIV is no longer a major underlining risk factor, since ARV therapy is broadly used, in Romania PcP is suspected in pediatric HIV-infected patients. In Serbia and Romania PcP diagnosis is made on clinical suspicion followed by empirical treatment. Only in Greece is molecular diagnosis of PcP available, together with microscopy and *P*. *jirovecii* DIF, but the number of referred pediatric patients is low. 

Type of specimen and laboratory methods play important roles in establishing PcP diagnosis. A possible explanation for the low number of examined pediatric patients could be the need for invasive respiratory sampling. For example, obtaining induced sputum or nasopharyngeal fluid is less invasive. However, depending on the technique and the experience of the laboratory, the sensitivity of these samples varies between 50% and 90% [28,29]. Moreover, in infants below 2 years of age, molecular assays performed on nasopharyngeal aspirate show positive results in only one third of cases [30]. BAL and open lung biopsy are most representative samples since they provide sensitivity of up to 90% or almost 100%, respectively [31,32]. However, these two sampling methods are highly invasive and thus, rarely performed in pediatric patients suspected for PcP.

However, another reason for the low demand in pediatric patients at risk is that they form a more homogenous and low-numbered group that are followed up by specific protocols and almost all of them are already under relevant prophylaxis, while this is not the case for the adult population.

Literature data regarding diagnostic methods showed that the most commonly used techniques for PcP diagnosis are microscopy of stained respiratory specimens from lung tissue. Disadvantages of these procedures are that they have low sensitivity, require trained staff, and are time consuming [33]. *P*. *jirovecii* DIF assay is more sensitive than conventional staining [34], but it can be applied only in a laboratory with fluorescent microscopy and experienced staff. Novel diagnostic options which overcome the limitations of previously-mentioned techniques for PcP diagnosis are molecular methods based on nucleic acid detection and quantification of *P*. *jirovecii* in respiratory specimens [35], as well as the serum-based pan-fungal BDG assay [36]. Less-expensive molecular assays for detection of *P. jirovecii*, such as loop-mediated isothermal amplification, is an alternative [37]. Recently, the next generation sequencing (NGS) and measurement of *P. jirovecii*-free DNA from peripheral blood showed promising results in immunocompromised hosts [38]. Molecular assays have increased detection rates of *P*. *jirovecii*, but the wide use of these methods is limited probably due to economic reasons, low incidence of PcP, and sometimes false positive results. *P. jirovecii* is a ubiquitous organism colonizing lungs in early age [6,39] and this could explain the “false” positive PCR results in these circumstances. Thus, interpretation of molecular assays is often difficult, mainly in how to distinguish colonization from infection. The combination of different diagnostic assays and different patient specimens with the application of global diagnostic guidelines for PcP [40,41] provides a possible solution to overcome the latter. Nowadays, many commercial techniques are available, so, facilitating implementation of molecular techniques like qPCR, together with staff training, are urgent in order to improve PcP diagnosis in pediatric patients.

This pilot work, as an initial effort to better understand the current status and diagnostic practices of PcP in pediatric hospitalized patients in the participating countries, showed that the clinical practice in terms of PcP prophylaxis, clinical suspicion, and treatment are similar in all three countries, while there are differences in laboratory diagnostic approaches. The challenge of improving availability of molecular diagnosis of PcP needs to be addressed. It might be done through the integrative laboratory platform which combines *P. jirovecii* DIF, *P. jirovecii* PCR, and pan-fungal BDG assay. Routine screening in larger numbers of samples would improve the knowledge and understanding of infection by *P. jirovecii*.

Regional collaboration could be promising. As PcP seems probably underdiagnosed in pediatric patients, an ongoing initiative to create an integrative diagnostic molecular perspective at the regional level, under the auspices of European Confederation for Medical Mycology (ECMM), for further molecular laboratory-based regional multinational investigations may cover the existing gaps and improve the knowledge and practices.

## Figures and Tables

**Table 1 jof-06-00049-t001:** Pediatric population (age 0–18) in Serbia (1,214,924), Greece (1,788,782), and Romania (4,170,598) in 2019: types of hospital unit, reported underlining diseases in pediatric population at risk, numbers of new cases at risk and current laboratory practices for *Pneumocystis* pneumonia (PcP)*.

Type of Hospital Units	Number of Relevant Units	Main Underlying Diseases	Estimation of New Cases of Underlying Disease Per Year (in Total)	Laboratory Reported Performing *Pneumocystis* Microscopy/DIF/PCR
**Serbia** PHO BMT HIV CF Organ transplant	5 1 1 1 1	Hematology/Oncology	~250	2/0/0
**Greece** PHO BMT HIV CF Organ transplant	7 1 2 2 In adult units	Hematology /Oncology	~320	**
**Romania** PHO BMT HIV CF Organ transplant	12 3 9 4 3	Hematology /Oncology HIV	~520 ~26	3/0/0

Abbreviations: DIF: direct immunofluorescence assay; PCR: polymerase chain reaction. Notes: * excluding preterm infants (data not obtained). ** In Greece, microscopy of stained slides is performed mainly in histopathology departments. Direct immunofluorescence (DIF) and PCR can be performed in several microbiology departments.

**Table 2 jof-06-00049-t002:** An example of available data concerning underlying diseases and laboratory approaches of 12 pediatric cases, clinically suspected for *Pneumocystis* pneumonia (PcP) and referred to a single reference laboratory during 2014 to 2019 (data of the First Department of Microbiology, Medical School, Aristotle University of Thessaloniki).

Number	Age (Years)	Gender	Underlying Disease	Chest Radiography	Laboratory Method Applied: Microscopy, DIF, PCR (Type of Biological Specimens)	Test Result Positive (+) or Negative (-)	Treatment for PcP Yes/No	Prophylaxis for *Pneumocystis* Yes/No
1	8	M	ALL	Yes	DIF (nasopharyngeal fluid)	(+)	Yes	Yes
2	7	M	ALL	Yes	DIF (nasopharyngeal fluid)	(-)	No	Yes
3	11	F	ALL	Yes	DIF (sputum)	(-)	No	Yes
4	2.5	F	ALL	Yes	DIF (sputum)	(-)	No	Yes
5	14	M	ALL	Yes	DIF (sputum)	(-)	No	Yes
6	15	F	Kidney transplantation	Yes	DIF (BAL)/PCR (BAL), repetition (x4) of DIF/PCR until negative result	(+)/(+) final: (-)/(-)	Yes	Yes
7	6 months	M	Mediastinal tumor/chemotherapy	Yes	DIF (BAL)	(-)	No	Yes
8	5.5 months	F	ALL, acute renal failure	Yes	DIF (BAL)	(-)	No	Yes
9	16	F	Kidney transplant	Yes	DIF (sputum)	(-)	No	Yes
10	7	M	ALL	Yes	DIF (sputum)	(-)	No	Yes
11	17	M	Kidney transplant	Yes	DIF/PCR (sputum) repetition of DIF/PCR	(+)/(+ ), second time again (+), under treatment	Yes	Yes
12	10	M	Sigmoid sinus thrombosis, Lymphadenopathy	Yes	DIF (sputum)	(-)	No	No

Abbreviations: M: male; F: female; ALL: acute lymphoblastic leukemia; DIF: direct immunofluorescence assay; PCR: polymerase chain reaction, BAL: bronchoalveolar lavage.

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
