# Peer review of "Diagnosis of Pneumocystis jirovecii Pneumonia in Pediatric Patients in Serbia, Greece, and Romania. Current Status and Challenges for Collaboration"

_jof, 2020, doi:10.3390/jof6020049_

Round 1

Reviewer 1 Report

The current version is much improved. 

Some changes that would in my view increase the clarity of the manuscript:

Title: Too long and the second sentence does not adequately reflect the contents.  Would propose:

Diagnosis of Pneumocystis jirovecii pneumonia in paediatric patients in Serbia, Greece and Romania. Current status and challenges for collaboration. 

Line 33: the sentence may read better as "Molecular diagnosis is available only in Greece."

Line 78: Suggest: iii) propose an integrative development of molecular diagnosis in the basis of increasing regional-scale collaboration. 

Line 113: Just a personal comment and reccomendation: There is a mith regarding resistance of P. jirovecii to TMP-SMX. The evidence for clinical resistence of P. jirovecii to TMP-SMX is the documented slow improvement of PCP in patients with mutations in their P. Jirovecii isolates. The efficacy of TMP is unquestionable. Therefore, I would prefer to delete "it was difficult to attribute the clinical improvement to TMP-SMX". 

Line 236: Reads: New perspectives and challenges based mainly on molecular PcP diagnostics are neccesary and important. 

Reccomend to replace with: The challenge of improving availability of molecular diagnosis of PcP needs to be addressed. 

Author Response

Dear Editor,

We would like to thank the reviewers for their time and the comments done. Please find below their remarks together with our relevant answers. The whole numbering of lines has been changed.

Point 1. Title: Too long and the second sentence does not adequately reflect the contents.  Would propose: Diagnosis of Pneumocystis jirovecii pneumonia in paediatric patients in Serbia, Greece and Romania. Current status and challenges for collaboration. 

Response 1. The new title is “Diagnosis of Pneumocystis jirovecii pneumonia in paediatric patients in Serbia, Greece and Romania. Current status and challenges for collaboration”

Point 2. Line 33: the sentence may read better as "Molecular diagnosis is available only in Greece."

Response 2. Line 33: Molecular diagnosis is available only in Greece.

Point 3. Line 78: Suggest: iii) propose an integrative development of molecular diagnosis in the basis of increasing regional-scale collaboration. 

Response 3. Line 79: iii) propose an integrative development of molecular diagnosis in the basis of increasing regional-scale collaboration. 

Point 4. Line 113: Just a personal comment and reccomendation: There is a mith regarding resistance of P. jirovecii to TMP-SMX. The evidence for clinical resistence of P. jirovecii to TMP-SMX is the documented slow improvement of PCP in patients with mutations in their P. Jirovecii isolates. The efficacy of TMP is unquestionable. Therefore, I would prefer to delete "it was difficult to attribute the clinical improvement to TMP-SMX". 

Response 4. We delete "it was difficult to attribute the clinical improvement to TMP-SMX". 

Point 5. Line 236: Reads: New perspectives and challenges based mainly on molecular PcP diagnostics are neccesary and important. Reccomend to replace with: The challenge of improving availability of molecular diagnosis of PcP needs to be addressed. 

Response 5. Line 240: We replace it with: The challenge of improving availability of molecular diagnosis of PcP needs to be addressed. 

Reviewer 2 Report

The manuscript has been significantly improved and reads very well. I have some few comments, which could be introduced in the Discussion.

Line 205 : induced sputum or nasopharyngeal fluid are commonly used for the diagnosis of PCP in developed countries, and yield a good sensitivity, provided that qPCR is used as diagnostic tool. Indeed, microscopy has poor sensitivity on these samples. The authors should stress the urgent need to implement qPCR techniques in their labs, to improve PCP diagnosis in pediatric patients. Nowadays, many commercial techniques are available, thus facilitating implementation and staff training.

Table 1: PCP is still responsible for RDS in preterm infants. This category is not reported in the table, does it mean that PCP is not diagnosed in these three countries in such patients ?

Author Response

Dear Editor,

We would like to thank the reviewers for their time and the comments done. Please find below their remarks together with our relevant answers. The whole numbering of lines has been changed.

Point 1. Line 205 : induced sputum or nasopharyngeal fluid are commonly used for the diagnosis of PCP in developed countries, and yield a good sensitivity, provided that qPCR is used as diagnostic tool. Indeed, microscopy has poor sensitivity on these samples. The authors should stress the urgent need to implement qPCR techniques in their labs, to improve PCP diagnosis in pediatric patients. Nowadays, many commercial techniques are available, thus facilitating implementation and staff training.

Response 1. Lines 234-236 Nowadays, many commercial techniques are available, so, facilitating implementation of molecular techniques like qPCR, together with staff training, are urgent in order to improve PcP diagnosis in paediatric patients.

Point 2. Table 1: PCP is still responsible for RDS in preterm infants. This category is not reported in the table, does it mean that PCP is not diagnosed in these three countries in such patients?

Response 2. Line 172: Notes: *excluding preterm infants-data not obtained.

This manuscript is a resubmission of an earlier submission. The following is a list of the peer review reports and author responses from that submission.

Round 1

Reviewer 1 Report

The authors titled their article « Molecular diagnosis of  Pneumocystis jirovecii pneumonia in pediatric patients”, but they do not present molecular results. It is difficult to class this paper: is it a survey, a review on practice or a review of literature ? The form is uncertain and confused (not a scientific writing, nor a review), and at first glance mostly resembles to an educational paper. The objectives are not clear, nor are the methods. Why melting a study on diagnostic practices (lines 84_90) with a  review (lines 92-93)? Among the few aims of the study announced (lines 77-82), only the first one is partially met, namely “collect available initial data on demographical  characteristics, underlying disease, …, treatment in pediatric patients with PcP”.

The results show only one Table which legend is very awkward : “examples of pediatric cases”: what does it mean exactly ? there is no data analysis. Do the author think that it is of any use for the readers?

Overall, the article presents the way the authors would enumerate the number of PcP cases in countries where the biological diagnosis is not made (at least in 2 out of 3 of them), which seems a difficult task, as clinical and radiological signs are not specific. Despite the fact that it is stated that there were differences in the management of patients between the three countries, “which would be important to be analyzed” (line 96), there is no analysis at all. At least a Table gathering the main findings, the number of centers involved in each country, the type of patients, the estimated incidence, a.s.o. would be welcome.  

The Discussion is long  and of little interest, as it is purely theoretical and disconnected from the few results shown. The § on molecular diagnosis (lines 199-210) is surrealist and suggests that the authors have no idea on what methods molecular diagnosis of PcP is based on. There is a confusion between diagnosis and genotyping methods. Let alone that NGS is not used for diagnosis and LAMP is anecdotally used… real-time PCR is not even cited !

Minor comments:

It is useless to state thrice that there is some initiative of structuring a future study “on the auspices of ECMM”.

Bibliography: Citations of Abstract book of TIMM is not allowed. Some references have heterogeneous typing.

Introduction and Discussion should not be parted in numbered sub-sections.

Author Response

Point 1: The authors titled their article « Molecular diagnosis of Pneumocystis jirovecii pneumonia in pediatric patients”, but they do not present molecular results.

Response 1: Thank you for this suggestion. The new title is “Diagnosis of Pneumocystis jirovecii pneumonia in pediatric patients - regional experiences, challenges and perspectives in molecular diagnosis”.

Point 2:  It is difficult to class this paper: is it a survey, a review on practice or a review of literature? The form is uncertain and confused (not a scientific writing, nor a review), and at first glance mostly resembles to an educational paper.

Response 2: Thank you for this suggestion. We improve our results in order to create the original article based on survey data. 

Point 3:  The objectives are not clear, nor are the methods. Why melting a study on diagnostic practices (lines 84_90) with a review (lines 92-93)? Among the few aims of the study announced (lines 77-82), only the first one is partially met, namely “collect available initial data on demographical characteristics, underlying disease, …, treatment in pediatric patients with PcP”.

 Response 3: Thank you for this suggestion. We modified objectives (lines 77_80)

Point 4: The results show only one Table which legend is very awkward: “examples of pediatric cases”: what does it mean exactly? there is no data analysis. Do the author think that it is of any use for the readers?

Response 4: Thank you for this suggestion. We modified section Results and Table 1 which is now Table 2 with new title (lines 159_164). We add new Table 1 (lines 144_149) containing number of center in each country, the type of patients, and the estimated incidence

Point 5: Overall, the article presents the way the authors would enumerate the number of PcP cases in countries where the biological diagnosis is not made (at least in 2 out of 3 of them), which seems a difficult task, as clinical and radiological signs are not specific. Despite the fact that it is stated that there were differences in the management of patients between the three countries, “which would be important to be analyzed” (line 96), there is no analysis at all. At least a Table gathering the main findings, the number of centers involved in each country, the type of patients, the estimated incidence, a.s.o. would be welcome

Response 5: Thank you for this suggestion. We modified results and discussion. Data analysis is done in session Discussion (lines 166_174, 183_186, and 208_212). We add new Table 1. (lines 144_149) containing number of center in each country, the type of patients, and the estimated incidence

 Point 6:   The Discussion is long and of little interest, as it is purely theoretical and disconnected from the few results shown. The § on molecular diagnosis (lines 199-210) is surrealist and suggests that the authors have no idea on what methods molecular diagnosis of PcP is based on. There is a confusion between diagnosis and genotyping methods. Let alone that NGS is not used for diagnosis and LAMP is anecdotally used… real-time PCR is not even cited

Response 6: Thank you for this suggestion. We modified the Discussion section (lines 166_227), and we added new reference for NGS (No 40)

 Point 7:   It is useless to state thrice that there is some initiative of structuring a future study “on the auspices of ECMM”.Bibliography: Citations of Abstract book of TIMM is not allowed. Some references have heterogeneous typing

Response 7: Thank you for this suggestion. We removed citations of Abstract book of TIMM

 Point 8:   Introduction and Discussion should not be parted in numbered sub-sectio

Response 8: Thank you for this suggestion. We modified it and put sub-section together 

Reviewer 2 Report

This questionnaire-based study reports cases of PcP in Serbia, Greece and Romania, and the state of the art diagnostic methods in use in these countries. The study is a first effort to understand frequency of Pneumocystis pneumonia and diagnostic practices in the participant countries. 

The introduction needs to update about the status of underlying causes of immunosuppression leading to PCP in children in the participant countries.  The manuscript does not refer to the AIDS epidemic, to their access to ARV therapy, nor to the frequency of other immunosuppressive conditions in general. For example, in line 62 page 2 the description is too general. Pediatric patients may have an abrupt course depending on the underlying cause. 

The manuscript needs improvement with a definition of colonization versus pneumonia, focus the manuscript and strongly needs to better outline the differences between the countries that is stated as the aim of the paper.  For example, a table comparing the participant countries and the number of PCP cases diagnosed and methods used in each country is needed.

Discussion 

The discussion, as presented is not clear on whether the authors intend a general review or a discussion of the results of their questionnaire. Therefore, the discussion should focus and aim first to comment the results of their questionnaire and then to briefly review the topic. 

References should be revised. For example, reference 38 corresponds to adults with PCP. It may be better to use reference in Clin Infect Dis https://doi.org/10.1093/cid/cis870 from the same group. 

Minor

Page 2-63 PcP may have a more...

Page 2-71 Normal chest CT scan findings alone do not rule out PCP: This sentence is misleading.  May be you refer to chest radiography.  Please document this statement better. 

Page 3-105 P. jirovecii (please use ii). 

Page 3-107 lavat (lavage?)

Page 5-172 inducted (induced?)

Author Response

Point 1: This questionnaire-based study reports cases of PcP in Serbia, Greece and Romania, and the state of the art diagnostic methods in use in these countries. The study is a first effort to understand frequency of Pneumocystis pneumonia and diagnostic practices in the participant countries

 Response 1: Thank you for this comment. We also added the paragraph with the limitation of the study (lines 228_232)

 Point 2: The introduction needs to update about the status of underlying causes of immunosuppression leading to PCP in children in the participant countries.  The manuscript does not refer to the AIDS epidemic, to their access to ARV therapy, nor to the frequency of other immunosuppressive conditions in general. For example, in line 62 page 2 the description is too general. Pediatric patients may have an abrupt course depending on the underlying cause. 

 Response 2: Thank you for this comment.

We added new Table 1 with key underlining diseases and estimated incidence. Also the number of centers per country added (lines 144_149)

Regarding ARV therapy we added information (line  139_141), and in secession Discussion line 170_171

 Point 3: The manuscript needs improvement with a definition of colonization versus pneumonia, focus the manuscript and strongly needs to better outline the differences between the countries that is stated as the aim of the paper.  For example, a table comparing the participant countries and the number of PCP cases diagnosed and methods used in each country is needed.

 Response 3: Thank you for this comment.

We added a definition of colonization versus pneumonia (lines 197_199)

The differences between countries showed in Table 1 (lines 144_149), and in Discussion (lines 169_173)

In Table 2 (lines 159_164) we added the only existing laboratory data from Greece, since there is no laboratory diagnosed PCP cases in Serbia and Romania

 Point 4: The discussion, as presented is not clear on whether the authors intend a general review or a discussion of the results of their questionnaire. Therefore, the discussion should focus and aim first to comment the results of their questionnaire and then to briefly review thetopic.

 Response 4: Thank you for this comment.

The Discussion session is now modified.

First we commented the results and then briefly review the topic.

 Point 5: References should be revised. For example, reference 38 corresponds to adults with PCP. It may be better to use reference in Clin Infect Dis https://doi.org/10.1093/cid/cis870 from the same group

 Response 5: Thank you for this comment.

We removed the reference 38, and added the reference doi.org/10.1093/cid/cis870 (No 37)

 Point 6: Minor

Page 2-63 PcP may have a more... Page 2-71 Normal chest CT scan findings alone do not rule out PCP: This sentence is misleading.  May be you refer to chest radiography.  Please document this statement better

Page 3-105 P. jirovecii (please use ii). 

Page 3-107 lavat (lavage?)

Page 5-172 inducted (induced?)

 Response 6: Thank you for this comment. We did this minor corrections

Reviewer 3 Report

The title seems not to be appropriate, as only one center made molecular diagnostic for PcP.

It’s not clear if the work aims to diagnose PcP in immunocompromised or immunocompetent children.

Nowadays, the gold standard to diagnose PcP is molecular techniques.

 1-     Concerning the aims of the work 

The first aim was to collect data about patients with suspected PcP, but in your introduction, PcP is described as non-specific symptoms, so the clinical diagnosis of PcP alone is not pertinent.

I agree that this is suspected PCP, as mostly no diagnosis of PcP is available (In Serbia, no biological diagnosis; in Romania, biological diagnostic relies on direct microscopy /immunofluorescence, which is insufficient in immunocompromised patients; in Greece diagnosis relies mostly on DIF, and even patient with negative test is included as suspected PCP).

The population described are all immunocompromised patients, they might be infected by so much others pathogens, that the title of the study would be "pulmonary infection in immunocompromised than PcP infection".

 The second aim was to discuss the type of patients samples and method of diagnosis. This part of the work should be presented in the results, and not in the discussion. These are no new information. It would be of interest to have a table with a comparison of the sensitivity of these techniques compare to the gold standard: molecular techniques on BAL.

In the second part of the discussion, the authors said that microscopic visualization is available in these 3 countries: Why in the description of Serbia patients, authors didn’t describe patients with a microscopic visualisation? And why authors include patients with negative DIF in their patient from Greece. I don’t agree with the term gold standard for microscopic visualization of PcP. It might be the only possible diagnostic in some countries/regions, but the gold standard still relies on molecular techniques.

 The third aim was « to prepare the platform for multinational PcP studies ». I’m not aware of this platform, and I didn’t find any information concerning the aim, the construction, the role,… of this platform, so I can not judge if this part of the work is described in this article

 2-     Concerning the construction of the manuscript

Each paragraph is to be rewritten, all information are mixed, without any order: for example: information concerning patients, haematological patients, HIV patients,immunocompetent, new born .... 

In your results part, the result of retrospective analysis from each country is mixed with the Pubmed research. They should be separate as it’s not the same part of the study.

Conclusions are not supported by the data.

The discussion seems to be not adapted to the rest of the article, authors described all new molecular techniques but these have not been used for their analysis Furthermore the molecular techniques described here, are not discussed regarding the aim of the work which was to diagnose PcP.

BDglucane can not be used as a diagnostic marker as its not specific 

Microbiome interactions seem to be out of context.

Conclusion is not in agreement with the content of the article 

Author Response

Point 1: The title seems not to be appropriate, as only one center made molecular diagnostic for PcP

 Response 1: Thank you for this suggestion. The new title is   “Diagnosis of Pneumocystis jirovecii pneumonia in pediatric patients - regional experiences, challenges and perspectives in molecular diagnosis”.

 Point 2: It’s not clear if the work aims to diagnose PcP in immunocompromised or immunocompetent children Nowadays, the gold standard to diagnose PcP is molecular techniques.

 Response 2: Thank you for this suggestion. We clarified that the aim is to diagnose PcP in immunocompromised children. We escape to use term “the gold standard”

 Point 3: Concerning the aims of the work  The first aim was to collect data about patients with suspected PcP, but in your introduction, PcP is described as non-specific symptoms, so the clinical diagnosis of PcP alone is not pertinen

 Response 3: Thank you for this suggestion. We made correction and modified aims (lines 77_80)

 Point 4:  I agree that this is suspected PCP, as mostly no diagnosis of PcP is available (In Serbia, no biological diagnosis; in Romania, biological diagnostic relies on direct microscopy /immunofluorescence, which is insufficient in immunocompromised patients; in Greece diagnosis relies mostly on DIF, and even patient with negative test is included as suspected PCP). The population described are all immunocompromised patients, they might be infected by so much others pathogens, that the title of the study would be "pulmonary infection in immunocompromised than PcP infection".

 Response 4: Thank you for this suggestion. We agree with your comment and we discussed it (lines 231_232). We wanted to point out the main limitation in etiological diagnosis of pulmonary infection and to focus on PcP which is most underdiagnosed and underestimated (lines 73_75)

 Point 5: The second aim was to discuss the type of patients samples and method of diagnosis. This part of the work should be presented in the results, and not in the discussion. These are no new information. It would be of interest to have a table with a comparison of the sensitivity of these techniques compare to the gold standard: molecular techniques on BAL.

 Response 5: Thank you for this suggestion. We modified section Results. In Table 1 we presented data from all three countries (lines 144_149). We modified Table 1, which is now Table 2 and put a new title (lines 159_164).

We discussed the sensitivity of these techniques and the representatives of the samples, including  BAL (lines 175_182)

 Point 6: In the second part of the discussion, the authors said that microscopic visualization is available in these 3 countries: Why in the description of Serbia patients, authors didn’t describe patients with a microscopic visualisation? And why authors include patients with negative DIF in their patient from Greece. I don’t agree with the term gold standard for microscopic visualization of PcP. It might be the only possible diagnostic in some countries/regions, but the gold standard still relies on molecular techniques.

 Response 6: Thank you for this suggestion. We discussed this and clarified that we did not received the BAL or other respiratory samples (lines 184_186). We removed the term gold standard for microscopic visualization of PcP.

 Point 7: The third aim was « to prepare the platform for multinational PcP studies ». I’m not aware of this platform, and I didn’t find any information concerning the aim, the construction, the role,… of this platform, so I can not judge if this part of the work is described in this article.

 Response 7: Thank you for this suggestion. We removed part from the text «to prepare the platform for multinational PcP studies».

 Point 8: Concerning the construction of the manuscript each paragraph is to be rewritten, all information are mixed, without any order: for example: information concerning patients, haematological patients, HIV patients, immunocompetent, new born. In your results part, the result of retrospective analysis from each country is mixed with the Pubmed research. They should be separate as it’s not the same part of the study.

 Response 8: Thank you for this suggestion. The discussion session is now modified. First we commented the results and then briefly review the topic.

 Point 9: Conclusions are not supported by the data.

 Response 9: Thank you for this suggestion. Data analysis and conclusions are done in session Discussion (lines 166_174, 183_186, 208_212, lines 228_240).

 Point 10: The discussion seems to be not adapted to the rest of the article, authors described all new molecular techniques but these have not been used for their analysis Furthermore the molecular techniques described here, are not discussed regarding the aim of the work which was to diagnose PcP.

 Response 10: Thank you for this suggestion. We adapted the Discussion in concordance with our results.

 Point 11: BDglucane cannot be used as a diagnostic marker as its not specific. 

Microbiome interactions seem to be out of context Conclusion is not in agreement with the content of the article.

 Response 11: Thank you for this suggestion. We added BDglucane as additional marker beside P. jirovecii DIF and P. jirovecii PCR. We removed the part “Microbiome interactions”. We rewritted the Conclusion (lines 166_174, 183_186, 208_212, lines 228_240).

Round 2

Reviewer 1 Report

What does Table 1 aim at ? an estimation of the annual incidence of PcP cases ? Then, another column should be added after the expected number of patients with underlying conditions. the labeling of the head columns is not clear. Please explain “Units type”. Does it refer to the medical units in charge of such patients? The number of hospitals would be informative, as well as the hospital type (tertiary care, secondary care…). And the possibility to diagnose PcP on site.

I don’t understand the sentence line 120-121.

What is the added value of Table 2? It shows that only 3 cases were confirmed and treated for PcP. What about the others? Were all patients under chemoprophylaxis before sampling ? Please explain more accurately the underlying message with these data.

Many English spelling or grammar errors, mainly in the modified sentences.

Reviewer 2 Report

Re-writing the introduction is a must. The aim of the manuscript is found after reading 75 lines of partly irrelevant introduction. The priority of the contents needs revision. Reaching line 75 is an effort that goes over information that is not relevant to the manuscript. Shortening can, also, be easily accomplished.  

This criticism may be better understood if the title is better focused: For example: "Current status and challenges for the diagnosis of Pneumocystis pneumonia in pediatric patients in Greece, Romania, and Serbia."

The abstract is better but writing needs to be more concise: For example on line 22: Diagnosis of PcP has traditionally relied on the visualization of the organism in respiratory specimens by microscopy. This method is available in our countries but has limited sensitivity. Molecular diagnosis is not yet widely implemented.  Phrases like "more or less" are imprecise and lack accuracy. 

Line 94: suggest: ...Greece. Relevant to this work, PcP is generally treated empirically and a consistent approach needs to be implemented. Regional experiences must be analyzed to adequately face the challenge of agreeing on an improved and common widely accessible diagnostic approach. 

Line 98: Suggest: Pediatric patients at high risk for PcP are...

Line 102: replace expect for except?

Table 1 underlining should be replaced by "underlying" 

Table 2: I dont understand the title of the 5th column "Lung Ro"

Discussion:

Should start witht the sentence: This retrospective study provides insight on the diagnostic practices and clinical approach to PcP in our region.